# Peptides, DNA and MIPs in Gas Sensing. From the Realization of the Sensors to Sample Analysis

**DOI:** 10.3390/s20164433

**Published:** 2020-08-08

**Authors:** Sara Gaggiotti, Flavio Della Pelle, Marcello Mascini, Angelo Cichelli, Dario Compagnone

**Affiliations:** 1Department of Medical, Oral and Biotechnological Sciences, “G. d’Annunzio” University Chieti-Pescara, via dei Vestini 31, 66100 Chieti, Italy; sgaggiotti@unite.it (S.G.); cichelli@unich.it (A.C.); 2Faculty of Bioscience and Technology for Food, Agriculture and Environment, University of Teramo, via Renato Balzarini 1, 64100 Teramo, Italy; fdellapelle@unite.it (F.D.P.); mmascini@unite.it (M.M.)

**Keywords:** gas sensors, sensor arrays, oligopeptide, molecularly imprinted polymers, oligonucleotide, E-nose, volatile organic compounds

## Abstract

Detection and monitoring of volatiles is a challenging and fascinating issue in environmental analysis, agriculture and food quality, process control in industry, as well as in ‘point of care’ diagnostics. Gas chromatographic approaches remain the reference method for the analysis of volatile organic compounds (VOCs); however, gas sensors (GSs), with their advantages of low cost and no or very little sample preparation, have become a reality. Gas sensors can be used singularly or in array format (e.g., e-noses); coupling data output with multivariate statical treatment allows un-target analysis of samples headspace. Within this frame, the use of new binding elements as recognition/interaction elements in gas sensing is a challenging hot-topic that allowed unexpected advancement. In this review, the latest development of gas sensors and gas sensor arrays, realized using peptides, molecularly imprinted polymers and DNA is reported. This work is focused on the description of the strategies used for the GSs development, the sensing elements function, the sensors array set-up, and the application in real cases.

## 1. Introduction

The discovery of genes encoding receptor ‘odour’ proteins, from the rhodopsin-like family, has been crucial to figure out how ‘smell’ works in humans. The odour perception mechanism at the gene and protein levels has been discussed by Buck [1]. Different authors have stated that an olfactory receptor (OR) can recognize numerous odorants with different affinity, thus, a single molecule can bind different receptor proteins [2,3,4]. More recently, Abaffy [5] discovered that odorous substances are able to bind receptor proteins, contained in olfactory glomeruli, that allow recognition. Humans have 851 olfactory receptor genes for the identification of thousands of different odours [6]. A complex receptor code, derived by the excitation of a set of olfactory neurons, results in perception, identification, detection of concentration changes as well as adaptation to particular odours. Each neuron has only one type of receptor. These studies have represented the starting point for the development of olfaction-inspired bioreceptors [7,8]. 

Advances in nano/micro-engineering and molecular biology have enabled the elaboration and development of devices that mimic the human olfactory system through the use of arrays of gas sensors (GSs) associated with signal-processing tools [2,3], the so-called E-noses. The e-nose concept started with the idea to mimic the olfactory system; thus, the basic set-up can be described using the following four elements: the sensors equipped with appropriate recognition elements (olfactory receptors), the transducers (bipolar neurons network), data elaboration (brain), and data interpretation (odour perception). The concept of an instrument consisting of an intelligent sensor array able of odour classification (model nose) was introduced by Persaud and Dodd [9]; the main challenges in the development of such devices were identified in finding the appropriate recognition (selectivity) and transducer (sensitivity) elements. An exhaustive definition of e-nose was given by Gardner and Bartlett [10]: ‘an electronic nose is an instrument, which comprises an array of electronic chemical sensors with partial specificity and an appropriate pattern-recognition system, capable of recognizing simple or complex odours’. In 2000, Göpel [4] reported on the problems generated by drifts in the signal and lack of sensitivity for certain classes of molecules. Improvements in more recent years have been characterized by progress in the development of new chemically sensitive materials, adaptable transducers, or by the combination of different sensitive materials and transducers in multielement modular sensor systems that may mimic the human odour panels. Furthermore, to reduce the drift effect in the GSs, new algorithms have been developed. Some approaches use multivariate analysis such as principal component analysis (PCA, see later) to compensate for drift [11,12,13]. Liu et al. [14], interestingly, used an active learning (AL) methodology, which allowed them to solve the sensor drift problem. This method can select a certain number of incoming samples and update the classifiers reliably. 

Different components of the biological olfactory system have been used as sensitive material to obtain GSs. The use of biological elements involved in the animals’ and humans olfactory receptors system has been reported in the literature, particularly odorant-binding proteins (OBP) [15,16], insect antennae [17] and olfactory neurons [18]. These represent the first obvious choice to mimic the binding ability for volatile compounds using the biological material selected by nature. On the other hand, it is worth to mention that other organic natural compounds as porphyrins, chemically synthesized and functionalized to have different binding ability within an array have been extensively and successfully used [19,20,21,22]. More recently, other binding elements have appeared in the gas sensors scenario, in particular for the development of gas sensors arrays.

Regardless of the transducer employed, the differences in the interaction of individual sensors with the same gas phase, maximized by a rational selection of the binding elements, represent a keystone for the success of the e-nose application. Therefore, a statistical data treatment needs to be carefully selected to interpret the GSs output, in particular for an array of sensors. The analysis of the outputs is generally directed to search for similarities and differences in the data set through a reduction in dimensionality. It is possible to use a classification approach grouping the information by discriminating features (i.e., inter-class distance); alternatively, explorative methods (i.e., variance) can associate the information obtained with the structure of the data. The appropriate statistical algorithm depends on the number of objects and their variables, the complexity of the problem, and the computational capabilities of the software [23]. Most of the statistical approaches for sensor arrays use multivariate data analysis, such as principal components analysis (PCA), discriminant analysis (DA), and clustering analysis (CA). PCA is a signal representation technique that generates projections along with the directions of maximum variance. DA is an explanatory and predictive analysis. Two different models of DA can be used to obtain different information: i) to which group an observation will belong to ii) if the groups to which the observations belong are distinct. A linear model can be used if the covariance matrices are assumed to be identical, while a quadratic model is used when the covariance matrices differ in at least two groups [24,25]. CA is used to split data into groups depending on the similarities or distances between points in the data set. Other methods that allow to study the results obtained from gas sensors are high-order statistics (HOS). These methods allow the extraction of more information than signal analysis and may lead to significant improvements in selectivity and sensitivity of sensor responses [26]. A detailed description of the proper use and different performances of these approaches are out of the scope of this review and is reported elsewhere, in particular regarding the output data interpretation of GSs arrays [27,28].

This review will focus on the realization and application of gas sensing systems recently developed using cutting edge binding elements as peptides, DNA, and molecularly imprinted polymers (MIPs). These binding elements are unique and useful tools for designing GSs resulting particularly appealing for sensors arrays set-up via molecular modeling, because of the ease of synthetic pathway and the low cost compared to the natural biological elements. Hence, they represent in our opinion, a new valid and alternative approach for the realization and boosting of GSs and e-noses, able to offer potential infinitive opportunities able to solve analytical and real-life challenges. Since GSs transducers represent a key element for the development of e-noses and GSs, and their modifications contribute significantly to the final GSs performance, in this review a brief description of the transducing systems used in conjunction with peptides, DNA and MIPs will be also given. Moreover, attention will be devoted even to immobilization, synthesis, and selection of the binding elements, as well as at the analytical application of the described devices.

## 2. Transducers

The relevant requirements of GSs and GSs arrays (e-noses) are, as for other chemical sensors, sensitivity, stability of the signal, repeatability of the output, reversibility (in terms of recovery time) [29]. As already reported in the introduction is very relevant to reduce sensor drift; in fact, the sensors are expected to work on real samples in the long term to build a robust multivariate statistical model (using real samples) to calibrate the array. Moreover, GSs that are modified with elements as peptides, DNA or MIPs are expected to work properly at temperatures not higher than 40 °C, to avoid denaturation/damage of the recognition elements. The material and structure, as well as the eventual modification (e.g. by using nanomaterials), of the surface of the transducer, is also crucial since a proper immobilization and orientation of the recognition element is necessary. A scheme of the main transducers used in conjunction with peptides, DNA and MIPs are sketched in Figure 1 and briefly described below: Field-effect transistors (FETs): FETs are devices controlling drain current by a gate voltage applied. They exhibit gas sensing ability when the metal gate is modified with proper material. The latter needs to electronically communicate with the gate and interact with the gas. The current/voltage variation can return information on the gas composition [30]. Metal oxide semiconductors (MOS) have been introduced in the e-nose technology to modify the FET’s gate. In MOSFETs, the threshold voltage of the sensor is sensitive to the interaction of certain gases on the gate material, usually a catalytic metal, because of the corresponding changes in the work functions of the metal and the oxide layers. The changes in the work functions are induced by the polarization of the surface and interface of the catalytic metal and oxide layers caused by the gas interacting with the catalytically active surface. Moreover, when the metal-insulator interface interacts with the gas, physical changes in the sensor occurs. Therefore, a porous gas-sensitive gate material is used to facilitate diffusion of gas into the material. Gas sensing MOSFETs are produced by standard microfabrication techniques, which incorporate the deposition of gas-sensitive catalytic metals onto the silicon dioxide gate layer [31,32]. Moreover, the FET gate modification with MOS allows the functionalization with biomolecules, allowing them to work at low temperatures [33]. However, another type of material can be used to modify the FET’s gate, graphene. It has been shown that these sensors can have different strengths such as high electrical conductivity, surface-volume ratio, low thermal resistance, and relatively low 1/f noise and the ability to strongly tune the conductivity of the gate. These aspects make these sensors promising for gas detection applications [34]; at the moment no application on peptides, MIPs or DNA sensors have been reported.Piezoelectric sensors: piezoelectric crystals can be used in different fields, including optoelectronics, electronics, liquid, and gas detection devices. Two kinds of piezoelectric sensors are used in gas sensing: surface acoustic wave (SAW) and quartz crystal microbalances (QCMs). The working principle is that a change in the mass of the piezoelectric sensor coating, caused by absorption of a volatile, gives a change in the resonant frequency. An input (transmitting) and output (receiving) interdigital transducer deposited on the top of the piezoelectric substrate are present in SAWs. The sensitive element is between the transducers, an acoustic two-dimensional wave propagates since potential is applied to the input transducer; frequencies are in the 100–400 MHz range [35,36]. Quartz crystal microbalances (QCM) are the most used piezoelectric sensors and are of great importance in fields as material science, environmental monitoring, electrochemistry, and biosensors. They are generally realized with an ‘AT cut’ quartz layer having 2 gold electrodes on each side; the crystal is forced to oscillate at the fundamental frequency using an alternating current [37]. When an external electric field is applied to the quartz, the frequency decay is proportional to the mass bound to the crystal [38]. The relationship between frequency change and film deposition efficiency that interacts with different VOCs is expressed by the Sauerbrey equation [39]. These sensors can be easily modified with biological elements using techniques such as drop-casting, spin coating and dip coating. The ease of realization, low costs, ability to work in real-time and the short analysis times make this type of transducers very attractive in the sensors field [40].Surface Plasmon Resonance (SPR): SPR is among the most used techniques for characterization and analysis molecular interactions particularly in biosensors; it has recently been used in gas sensing [41,42,43]. SPR is induced by the resonant coupling of photons from polarized light to the oscillation of metal-free electrons; this produces an evanescent electromagnetic wave through to the surface of the metal [44,45]. The binding of a target analyte to a bioreceptor on the sensor surface influences the wave and can be monitored through the variation of the angle of the reflected light onto an appropriate SPR sensor. The sensitivity of the SPR depends on the sensor configuration and particularly on the functionalization of the sensor surface [41]. Recent applications of surface plasmonic waves include surface plasmon-enhanced Raman scattering (SERS) [46,47], localized SPR (LSPR) [48,49,50], surface plasmon field-enhanced fluorescence spectroscopy (SPFS) [51]. The most used configuration for the SPR sensors is the Kretschmann configuration ATR coupling introduced by Kretschmann and Raether after the pioneering work of Otto in 1968 [52]. This configuration is also used as an excitation method in current SPR imaging (SPRi) sensors [53]. SPRi has been largely used for the development of biochips for monitoring biomolecular binding events. Brenet et al., have demonstrated for the first time that SPRi is very efficient for the development of e-noses for sensing VOCs in the gas phase [54]. Using imaging technology, it is possible, in principle, to simultaneously monitor hundreds of biofunctionalized spots in a micro-array format on the surface of the entire biochip [54].

## 3. Peptides, DNA and MIPs as Sensing Elements 

Technical solutions to improve the sensitivity and selectivity of artificial noses have been recently extensively reviewed by Hurot et al. [55]. In this excellent work, the authors give an overview of e-noses dedicated to the detection of volatile organic compounds (VOCs) and to biomimetic strategies to improve the design of the detection materials, their immobilization on the sensor surface, the sampling strategies, and data processing. In this review we report complementary information on the development of GSs based on oligopeptides, DNA and MIPs, focusing on their assembly and setup strategies, illustrating performances, target volatiles compounds, and applications, when reported, in real samples. 

The sensing elements in GSs are supposed to give a certain degree of selectivity versus the target compounds, as for classical bio-sensors working in liquid media; at the same time, they should be independent of physico-chemical changes occurring during the measurement (mainly temperature and humidity). 

A key element that differs from classical liquid phase affinity sensors is related to the reversibility of the signal; in fact, the signal should be rapid and reproducible, and no regeneration procedure should be required. Moreover, the immobilization on the transducer should be carefully selected to maximize the sensing element exposure. Sensing molecules can be immobilized directly on the transducer or onto supports that can then be fixed onto the transducer. Parameters affecting immobilization and the response are molecule size, working area, polarity, shape, presence of functional groups, orientation after immobilization, and storage conditions [5]. 

The sensing elements oligopeptides, MIPs, and DNA used for gas sensing are treated in the following paragraphs, reporting initially cases of GSs and arrays where standards volatiles are analyzed focusing later on the performances on real samples. A table (Table 1, Table 2 and Table 3) summarizing the main features of the sensors in terms of the binding element, analyte/sample, transducer, immobilization strategies, VOCs detected amount is reported at the end of each paragraph. 

### 3.1. Peptides 

Peptides are ideal to mimic the molecular recognition mechanism occurring in biomolecules such as enzymes, antibodies, receptors (including olfactory) and transmembrane proteins. The design of peptide-based artificial receptors able of highly specific recognition is widely used to obtain inhibitors, drugs, reagents for affinity purification systems, and, also, active elements for gas sensing [55]. In particular, short peptides represent an excellent opportunity for the design of artificial receptors, because of the impressive number of different combinations that can be obtained using the 20 natural amino acids. Figure 2 reports examples of oligopeptides structures.

Easy and automated synthesis, low cost and the possibility of rapid screening in virtual libraries [56,57] are also relevant issues for their selection. Peptide sequences for VOCs detection were initially developed in the early 2000s based on animal receptors. The first olfactory receptors-based peptide sequences were designed to maintain the olfactory receptor regions of dogs for VOC binding [15]. For the sake of clarity, the peptides reported below will be numbered using the symbol ‘Ps’ and consecutive numbers, the corresponding aminoacidic composition is reported in the ‘abbreviation’ list. The obvious choice to obtain a peptide for a gas sensor is to start from sequences already present in olphactory receptors. In this respect, Panigrahi et al. [58] studied the ability of the octapeptide Ps1 to detect acetic acid at low concentrations starting from the primary structure of the receptor protein P 30953 (from the Swiss prot G databank). After simulating to assess the binding ability to acetic acid, Ps1 was immobilized onto 10 MHz QCMs. Covalent immobilization was achieved by the formation of a self-assembled monolayer (SAM) onto the gold electrode of the QCM taking advantage of the presence of the thiol group of the cysteinyl end residue. This is a standard strategy to obtain a stable monolayer immobilization of biomolecules on gold. A linear increase of the response was observed in the 10–100 ppm range with a sensitivity slightly higher than 1 Hz/ppm. No data on selectivity are reported despite the authors stated that a signal was achieved also in presence of alcohols and ketones. 

Wasilewski et al. [59] used peptides mimicking the aldehyde binding regions in a hydrophobic cavity of the HarmOBP7 protein, identified as a pheromone-binding protein (PBP), present in the antennae of the Helicoverpa armigera moth. In the first paper [60] the binding ability of the peptide Ps2, constituted by 17 amino acids, versus 15 compounds (listed in Table 1) was predicted computationally. Ps2, thanks to a final cysteine, was anchored on the QCMs transducer by SAM. Maximum sensitivity was achieved for nonanal, helional and octanal with a frequency shift of 1.29, 0.77, and 0.69 Hz/ppm, respectively. No interactions were found for 100 ppm of formaldehyde, acetaldehyde, glyoxal, benzaldehyde, propanal, and hexanal. The in silico data correlated well with experimental results. Different shorter peptides Ps3, Ps4, Ps5, and Ps6 coming from the same HarmOBP7 receptor were also used to realize similar peptide GSs that were challenged against different odorous compounds (octanal, acetaldehyde, benzaldehyde, ethanol, acetone, dimethyl sulphide, trimethyl amine, and toluene). Adsorption of octanal on Ps6 was more favorable than for other sensors. However, a major drawback resulted in the long adsorption/desorption time which led to long times for the recovery of the signal. The Ps6 based sensor was the only returning significant signals, but for amounts of VOCs in the order of magnitude of thousands of ppm.

Lu et al. [61] in 2009 used a 10 MHz QCMs array of sensors based on four short synthetic oligopeptides and two conducting polymers for the simultaneous detection and identification of VOCs such as acetic acid, butyric acid, ammonia, dimethylamine, benzene, chlorobenzene, and their mixtures. The four polypeptide sequences Ps7, Ps8, Ps9, and Ps10 were from the dog olfactory protein OLFD-canfa’s (Swiss Prot G databank); spin coating was used to immobilize the peptides onto QCMs. The two conducting polymers were: monobenzo-15-crown-5 (B15C5) and poly [n-butyl methacrylate (PBMA)]. The amino acid sequences were selected according to the calculation of the binding affinity using virtual screening. The sensitivity of the array was tested with butyric acid. Ps7 and Ps8 had higher absorption towards butyric acid with respect to the conductive polymers. The sensitivity of the response of the two polypeptides towards various concentrations of butyric acid (1–3 ppm) in terms of slope of the response curve had the following sequence: Ps8 > Ps7 > Cp1, demonstrating that polypeptides have higher sensitivity than conductive polymers, as expected. Interestingly, the sensor array demonstrated selectivity for the discrimination of odour profiles characterized by all the VOCs tested.

An array of GSs based on the structurally related series of the well-known tripeptide glutathione (Ps11, Ps12, Ps13, Ps14, Ps15, Ps16) was developed by Compagnone et al. [62]. The aim was to exploit the ability of similar peptidic structures to discriminate among different VOCs, without any reference to olfactory receptors. Considering the very low response obtained by immobilization via SAM onto 20 MHz QCMs, the peptides were anchored to gold nanoparticles (AuNPs) that were then drop-casted onto the sensors. This increased the number of binding sites because of the large surface/volume ratio of the AuNPs. Typical aromas of foods as *cis*-3-hexenol, isopentyl acetate, ethyl acetate, and terpinen-4-ol dissolved in different solvents were tested. The resulting sensitivity pattern for all the peptides indicated different sensing abilities despite a similar structure. The intraday RSD was in the 1–10% range and interday RSD was in the 3–16% range for all the peptide QCM sensors.

The feasibility of SPR for detection in the gaseous phase was demonstrated in the early 1980s [63,64]. In 2018, Brenet et al. [54] developed an optoelectronic nose using peptides onto microarrays of a surface plasmon resonance imaging (SPRi) system. SPRi allows the immobilization of up to hundreds of sensing molecules on the same chip for the creation of a large sensor array. SPRi technology allows measurements using a fixed refraction angle called the working angle θw. Using an angle of incidence at the maximum slope of the plasmon curves, changes in resonance conditions can produce large variations in the intensity of reflectivity. In this system, the authors used a 16-bit video camera (CCD) to capture the interaction images between the VOCs and the sensor array. 18 different peptides and organic molecules such as thiols with diverse physicochemical properties (hydrophobic, hydrophilic, charged, neutral, etc.) were used as sensing materials. VOCs from different chemical families having different properties and distinct smells (alcohols, esters, carboxylic acids, ketones, hydrocarbons, aldehydes, and amines) were tested. The array was able to differentiate among VOCs of similar molecular structure. Two sets of homologous VOCs were analyzed (six alcohols with carbon chain lengths from C_3_ to C_8_, and five carboxylic acids from C_1_ to C_6_). PCA and Hierarchical Clustering on Principal Components (HCPC) were used to elaborate the dataset. Using PCA a good separation, particularly for alcohols, was achieved. The peptide sequences were not given for confidentiality reasons. 

Maho et al. [65] recently used this new technology to study the enantioselectivity of (*R*) and (*S*) limonene and (R) and (S) carvone. 19 different sensing materials have been deposited on the surface of the prism, 17 of them being peptides. The two remaining sensing materials were achiral molecules and were used as a control for the study. An automatic gas sampling system was used to create data sets (≥100 samples/class). Different lines were used for ambient air, for chiral forms of carvone and limonene, and butanol as control. The array was able to discriminate limonene and carvone enantiomers. A specific algorithm has been also proposed to assess which are the sensors more relevant for the discrimination. Moreover, Weerakkody et al., [66] and Brenet et al., [66] highlighted the importance of characterization of the optical system in the SPRi before depositing the array on the prism.

One of the attractive features of peptides is the possibility to obtain a particular binding ability designing a sequence by molecular modeling. Using this concept and starting from a sequence in the P 30953 receptor protein (Swiss prot G databank) in 2011, Sankaran et al. [67] obtained the synthetic peptide Ps17 designed for alcohols. Ps17 was immobilized onto 10 MHz QCMs by SAM and tested with hexanol and pentanol. A linear range from 2 to 100 ppm was obtained for both VOCs with a sensitivity of 0.05 Hz/ppm for hexanol and 0.03 Hz/ppm for pentanol, respectively. Despite no selectivity data are reported in the paper the data demonstrated that a peptide sequence can be designed to bind selected VOCs.

Sim et al. [68] very recently, reported a novel carbon nanotubes (CNT) FET functionalized with 12-mer peptides identified using a virtual screening approach (the peptide sequences are not reported). The authors used four different peptides to functionalize the SWCNTs without additional covalent conjugation linkers or surface modifications. Four breath-related VOC biomolecules—isopropyl alcohol (IPA), acetone, isoprene, and toluene were tested. Fabrication of the CNT FETs was achieved using peptide-functionalized CNT suspensions (1 ppm); the peptide-functionalized CNT-FETs were exposed to VOCs (10 ppm). The outputs of each sensor were analysed in terms of magnitude and response time of the resistance change. The resistance levels varied depending on the peptide sequence. The 4 peptide-functionalized CNT FETs had different responses for each VOC. 

Mascini et al. [69,70,71,72] developed a semi-combinatorial virtual approach to have a library of peptides potentially useful for GS arrays. The peptide library was built challenging the binding properties towards five different chemical classes (alcohols, aldehydes, esters, hydrocarbons, and ketones). By maximizing the differences of binding affinity among chemical classes, a subset of 120 tripeptides was used as scaffolds for generating a combinatorial library of 7912 tetrapeptides. Different peptides were selected depending on their virtual affinity and cross-reactivity to validate experimentally the approach (Ps18, Ps19, Ps20, Ps21, Ps22, Ps23). The peptides, added with a terminal cysteine, were covalently bound to AuNPs and deposited onto 20 MHz QCMs [62]. The array was able to discriminate 13 volatile compounds (2-propanol, ethanol, hex-3-en-1-ol, terpinen-4-ol, nonanal, octanal, ethyl acetate, ethyl butanoate, ethyl octanoate, isopentyl acetate, hexane, acetone, and butane-2,3-dione) on the base of hydrophobic/hydrophilic nature and molecular weight. The Ps18 designed, according to the simulation, for compounds having long alkyl chain with carbonyl group, had the highest response for all compounds followed by Ps22 except for 2-propanol and butane-2,3-dione for which Ps21 had a slightly higher response. The lowest response was in all cases given by the Ps19 that was designed for the recognition of small alcohols. The data obtained confirmed the efficacy of the virtual approach. Similar considerations applied for the same set of peptides immobilized onto zinc oxide nanoparticles (ZnONPs) [70]. The use of ZnONPs to immobilize the peptides resulted in a more stable background signal and in the absence of drift in presence of samples with high water content. 

The use of a library of peptides to realize GSs has been also reported by Ju et al. [73]. Ps24, Ps25, Ps26 generated by an ‘m13 phage display’ library, were selected for the binding of benzene, toluene, and xylene. The detector was, in this case, a microcantilever system consisting of four compartments, including three cantilevers each. Each compartment was independently functionalized with a peptide via SAM; the fourth was used as reference. Upon binding of the VOCs the resonant frequency of the signaling cantilevers downshifted more than the reference as a result of the selectively bound target molecules; the differential signal between the two cantilevers was expected to correspond to the concentration of target molecules in the gas sample. Ps24 was very selective toward benzene over toluene and xylene. On the contrary, Ps25 did not show specific binding toward benzene but had a high affinity toward toluene and xylene. Ps26 had significant interactions with benzene and toluene, but not with xylene. Ps24 was able to detect benzene down to 0.1 ppm, while Ps25 was quantitatively detecting toluene and xylene at 2 and 28 ppm. 

#### Applications in Real Samples

In 2014 Di Natale et al. [74] reviewed the development of gas sensors devoted to ‘breath analysis’. Breath is a widely challenging studied field in gas sensing because the breath sample collection is non-invasive and then, it is potentially considered a point of care testing. The gas sensors may be considered as the natural complement to breath analysis, matching the non-invasiveness with typical sensor features such as low cost, ease of use, portability, and integration with the information networks. This will potentially allow the screening of large populations for the early diagnosis of pathologies. 

To the best of our knowledge, only two examples of the use of gas sensor arrays based on peptides as medical diagnostic tool have been proposed. One of the first studies was reported in 2001 by Lin et al. [75] for the diagnosis of uremia. Six 12 MHz QCMs were dip-coated with peptides (sequences are not reported) designed by simulating the olfactory receptor protein (ORP) docking with the target gas molecules. A simulated structure of the ORP was modeled by comparing the primary sequences with protein G coupled receptor, which have known structures in the Protein data bank. The target molecules were dimethylamine (DMA), trimethylamine (TMA), monomethylamine (MMA), and ammonia, which are commonly found in the breath of patients with uremia. In a clinical test, the exhaled breath of patients in hemodialysis with chronic renal insufficiency (CRI), chronic renal failure (CRF) were collected and stored in sampling bags for the assay. The GSs array allowed clear discrimination between the samples from healthy subjects and subjects with chronic renal insufficiency/chronic renal failure. 

Ninety-six patients in an intensive care unit were tested by Shih et al., [76] using an array of peptide-based sensors. The sensor array was based on piezoelectric 10 MHz QCMs equipped with 24 different peptide chains grouped in hydrophilic and acidic, hydrophilic and basic, hydrophobic and acidic and hydrophobic and basic; each group was composed by 6 peptide sequences (sequences were not reported). The analysis of 128 breath samples was compared with bacteria culturing of saliva samples. Very interestingly six different bacterial pathogens (*Pseudomonas aeruginosa, Acinetobacter baumannii, Klebsiella pneumoniae, Staphylococcus aureus* and *Acinetobacter lwoffii*) were identified and grouped into clusters by multiple discriminant analysis (MDA); that is a multivariate dimensionality reduction technique. This method allows objects to be allocated consistently and most appropriately to groups for which representatives have already been selected [77]. The GSs array allowed a classification with 98% accuracy. 

The assessment of the quality of agricultural and food products is still routinely judged based on consumer preferences by visual inspection and subjective satisfaction levels. E-noses technology provides new options for assessing non-destructively the quality of agricultural and food products. This is particularly true when analysis should be compared with conventional methods such as GC-MS in which sample preparation and analysis requires time and skilled personnel [78]. Recently, peptide modified gas sensors arrays have been used for the evaluation of the aromas of different food matrices at the University of Teramo. Starting from the virtual library approach reported in [69,70,71,72] the pentapeptides developed have been used to assess quality, to classify different products, and to evaluate processes occurring in food. 

Using peptide-AuNPs functionalized with the glutathione series reported in [62] and adding a heptapeptide to improve variability in binding ability, Compagnone et al. [72] classified different commercial categories of olive oils as extra virgin, virgin, and lampante. With a similar approach, the performance of an array consisting of 6 peptide sensors (Ps15, Ps12, Ps27, Ps28, Ps29, Ps30, Ps27) has been compared with a metalloporphyrin based array for the evaluation of chocolate off-flavours potentially occurring during storage, transport and production. Off-flavoured samples of cocoa butter containing 125 ppm of 3-methylbutanal, phenylacetaldehyde (typical by-products of fermentation), acetic acid (produced in conching), tetramethylpyrazine, 2- acetylpyrrole (roasting), 2-nonenal and 2,4-decadienal (fat oxidation) have been prepared in the lab and tested [79]. A multivariate PLS-DA analysis was made to discriminate acceptable vs. off-flavour samples of dark, milk, and white chocolate. The peptides-based sensors performed better than the metalloporphyrin based array assigning correctly 98% of the samples (vs. 70%) demonstrating that this e-nose can be used in quality control of the samples at industrial scale.

The same sensor array was used to investigate the influence of the composition on the release and sensory perception of aroma compounds from strawberry flavoured candy model systems by Pizzoni et al., [80]. Gummy candies were prepared according to industrial manufacturing procedures using different gelling agents: gelatin, Arabic gum and pectin. Two types of strawberry flavours, natural and nature-identical (chemically synthesized) were used. The results obtained from GC-MS analysis showed a quite different and significant effect in aroma release due to the gelling agent. The peptide-based e-nose was able to discriminate the different gelling agents. More interestingly, for gelatin and Arabic gum strawberry candies, it was possible to clearly distinguish the natural from the chemically synthesized flavour, using a simple PCA. A panel test analysis was not able to give the same output. 

Rocchi et al. [81] used this type of GSs array to characterize and discriminate origin, drying, and age of 35 saffron samples. The pentapeptides (Ps18, Ps19, Ps20, Ps21, Ps22, Ps23) were tested with 35 saffron samples of different origin collected during the harvesting period 2012–2016. The identification of the volatile profile of saffron was achieved using GC-MS analysis. A PLS-DA analysis showed good discrimination by the GSs peptides array based on the origin of saffron samples with 81% of the samples correctly assigned in cross-validation.

AuNPs have been replaced by ZnONPs in [70] because of the improved performance in samples containing a high amount of water. As a proof of concept, an array of four different pentapeptides (Ps18, Ps19, Ps20, and Ps22) was tested on different samples of apricot, pear, and peach fruit juice. The same array was challenged with thermally sterilized carrots [82]. Sterilized samples were stored at three different temperatures: 25 °C, 4 °C and −18 °C. The signals were very stable and reproducible (inter-day RSD ≤ 15). The data outputs, elaborated by PCA, demonstrated that the array can monitor the evolution of the VOCs in carrots. 

A further application reported on food has been the evaluation of the volatile profile of pasta samples [83]. The final quality of pasta is strongly dependent on the quality of semolina and the drying step. The thermal damage of pasta is generally assessed by measuring, using HPLC, the extent of Maillard reaction and, in particular, by the amount of furosine, an Amadori compound formed during the early stage of the Maillard reaction. Twenty-seven samples of commercial pasta of different prices and different brands were analysed. GC-MS analysis of the headspace was evaluated by PCA. The score plot clearly distinguished between the low-price pasta (below 1 €/Kg) and high-price pasta samples. This separation can be associated with some aldehydes, alcohols, lactamide, and only one furanic compound (2-pentylfuran). Some compounds, markers of oxidative processes of lipids (indicating the low quality of the starting product) were detected only in low price pasta samples. The peptide-based e-nose was able to discriminate clearly among high and low/price pasta, indicating a possible use for the control of the quality of the semolina used. 

A new technology developed by Aribal Technologies and CEA-LETI Laboratory from Grenoble has been recently used to create an optoelectronic nose coupled to a miniaturized silicon pre-concentration unit; this was applied to the discrimination of volatile organic compounds emitted by flavored waters [84]. The pre-concentration unit coupled to the SPRi based e-nose was filled with around 7 mg of Tenax-TA adsorbent (mesh 80–100) and a thermo-resistive Ti/Pt film heater allowing the quick heating to 200 °C for the thermal desorption of the trapped VOCs. Nonane was used as a VOC model to characterize the signal amplification capacity of the pre-concentration unit. The headspace of flavored waters: grapefruit, lemonade, lemon, white peach, organic strawberry, strawberry, organic apple, mango and passion fruit were then tested. Before the thermo-desorption measurement, samples are disconnected, and the unit purged with ambient air. Total desorbed VOCs concentration of headspaces was measured with a photoionization detector. Only grapefruit (111 ppm) and organic apple (67 ppm) had enhanced different levels of VOCs, the other samples gave equivalent signals (10–17 ppm). PCA confirmed the results, grapefruit and organic apple samples were separated from the others. 

An overview of the different peptide-based gas sensors’ main features and applications are reported in Table 1.

**Table 1 sensors-20-04433-t001:** Gas sensors equipped with peptides sequences: main features and strategies.

Peptides	Analyte	Transducer	Immobilization Technique	VOCs Concentration Measured	Application	Ref
Ps1	Acetic acid	QCM	Self-assembly monolayer	10 ppm	-	[58]
Ps3, Ps4, Ps5, Ps6	Octanal, acetaldehyde, benzaldehyde, ethanol, acetone, dimethyl sulphide, trimethyl amine, and toluene	QCM	Self-assembly monolayer	Octanal 1435 ppm; benzaldehyde 2198 ppm; trimethyl amine 1594 ppm; acetaldehyde 4007 ppm; acetone 3028 ppm	-	[59]
Ps2	Aliphatic aldehydes, formaldehyde, acetaldehyde, propanal, pentanal, hexanal, heptaldehyde, octanal, nonanal, decanal, undecanal. dialdehyde-glyoxal; aromatic aldehydes-benzaldehyde, p-tolualdehyde, panisaldehyde, helional.	QCM	Self-assembly monolayer	>100 ppm	-	[60]
Ps7, Ps8, Ps9, Ps10	Acetic acid, butyric acid, ammonia, dimethylamine, benzene, chlorobenzene, and their mixtures	QCM	Spin coating	-	-	[61]
Ps11, Ps12, Ps13, Ps14, Ps15, Ps16	*cis*-3-Hexenol, isopentyl acetate, ethyl acetate, terpinen-4-ol	QCM	Self-assembly monolayer	-	-	[62]
-	Alcohols, esters, carboxylic acids, ketones, hydrocarbons, aldehydes, and amines	SPRi	Micro spotting robot	2-methylpyrazine 290 ppm; phenol 34 ppm; isoamyl butyrate 70 ppm; 1-pentanoic acid 51 ppm; 1-pentanol 47 ppm; and 1-octanol 8 ppm	-	[54]
-	(*R*) and (*S*) limonene; (*R*) and (*S*) carvone	SPRi	Micro spotting robot	-		[65]
Ps17	Hexanol and pentanol	QCM	Self-assembly monolayer	Hexanol 2–3 ppm; pentanol 3–5 ppm	-	[67]
-	IPA, acetone, isoprene, toluene	SWCTs-FET	-	10 ppm	Breath tests	[68]
Ps18, Ps19, Ps20, Ps21, Ps22, Ps23	2-Propanol, ethanol, hex-3-en-1-ol, terpinen-4-ol, nonanal, octanal, ethyl acetate, ethyl butanoate, ethyl octanoate, isopentyl acetate, hexane, acetone, butane-2,3-dione	QCM	Drop casting	-	-	[69,70,72]
Ps24, Ps25, Ps26	Benzene, toluene, and xylene	Cantilever array	Self-assembly monolayer	Benzene 0.012 ppm toluene 2 ppm xylene 28 ppm	-	[73]
-	Dimethylamine, trimethylamine, monomethylamine, and ammonia	QCM	Dip-coated	-	Breath tests	[75]
-	-	QCM		-	Bacterial infection: *Pseudomonas aeruginosa*, *Acinetobacter baumannii*, *Klebsiella pneumoniae*, *Staphylococcus aureus*, and *Acinetobacter lwoffii*	[76]
Ps11, Ps12, Ps13, Ps14, Ps15, Ps16	*cis*-3-Hexenol, isopentyl acetate, ethyl acetate, terpinen-4-ol-	QCM	Self-assembly monolayer	-	Olive oil	[62,72]
Ps15, Ps12, Ps27, Ps28, Ps29, Ps30, Ps27	3-Methylbutanal, phenylacetaldehyde, acetic acid tetramethyl-pyrazine, 2-acetyl-pyrrole, 2-nonenal and 2,4-decadienal	QCM	Self-assembly monolayer	-	Dark, milk, and white chocolate	[79]
Ps15, Ps12, Ps27, Ps28, Ps29, Ps30, Ps27	-	QCM	Self-assembly monolayer	-	Gummy candies	[79]
Ps18, Ps19, Ps20, Ps21, Ps22, Ps23	-	QCM	Drop casting	-	Saffron	[81]
Ps18, Ps19, Ps20, and Ps22	-	QCM	Drop casting	-	Fruit juice	[70]
Ps18, Ps19, Ps20, Ps21, Ps22, Ps23	-	QCM	Drop casting	-	Carrots	[82]
Ps18, Ps19, Ps20, Ps21, Ps22, Ps23	-	QCM	Drop casting	-	Pasta	[83]
-	Nonane	SPRi	Micro spotting robot	10–111 ppm	Flavored waters	[84]

### 3.2. MIPs

Molecularly imprinted polymers (MIPs) are cross-linked synthetic materials with artificially generated recognition sites able to selectively bind a target molecule [85]. Molecular imprinting involves the preparation of a “mold”, that is achieved using monomers that are polymerized around the target compound. After polymerization, the target molecule is removed by extensive and specific washing steps to leave recognition sites that are supposed to be complementary to the model molecule in terms of size, shape, and position of the functional groups. The synthesis of MIPs is relatively cheap and simple: basically, it requires the target molecule, one or more functional monomers, a cross-linker, an initiator, a porogenic solvent, and a solvent for mold extraction after complex polymerization. Different strategies for the realization of the MIPs can be carried out depending on the nature of the target molecules and the envisaged use of the MIPs within the assay format (i.e., for solid-phase extraction or sensor surface modification). These approaches and the use and classification of MIPs (often reported as ‘plastic antibodies’) have been extensively studied and reviewed in the literature [86,87,88]. Figure 3 shows a graphical schematization of the MIPs assembling.

Feng et al. [89] in 2005 reported the first use of a MIP in GS for the detection of the toxic gaseous formaldehyde. The MIP was prepared using methacrylic acid (MAA) as functional monomer, ethylene glycol dimethacrylate (EGDMA) as crosslinker, and 2,2-azobis (2,4-dimethyl) valeronitrile as initiator. The mixture was polymerized under UV light onto one side of 9 MHz QCMs. As usual for MIPs, nonimprinted polymers (NIPs) were also synthesized as control. Saturation of the response was achieved at ~2 ppm. The selectivity of the MIP for formaldehyde was confirmed using other similar VOCs (benzaldehyde, acetone, acetic ether, and ether) and comparing the response with the NIP GSs. Another work for the detection of formaldehyde has been reported more recently. Hussain et al. [90] realised a different MIP QCMs introducing styrene into the MAA polymer to reduce the polarity. The polymerization occurred by UV irradiation after the spin-coating of the oligomer solution onto 10 MHz QCMs. The frequency shift was selectively dependent on formaldehyde concentration in the 1–100 ppm range. 

Matsuguchi et al. [91], interestingly, prepared MIPs based on methyl methacrylate (MMA) in the presence of toluene and *p*-xylene as porogens and evaluated the molecular imprinting ability vs. the solvents. Divinylbenzene (DVB) was the cross-linking agent and benzoyl peroxide the radical initiator. The MIPs were deposited by spin-coating onto 4 MHz QCMs before polymerization. The sensor response was measured at 540 ppm for toluene and 170 ppm for *p*-xylene vapours. The change in the frequency of the crystal caused by the VOC vapor sorption was in accordance with the Sauerbrey equation. Using this approach selectivity of the MIPs-QCM sensors for the porogen solvent used to synthesize the polymer was achieved.

Terpenes are an important class of plant constituents deriving from different combinations of C5 isoprene subunits. They are known to possess various medicinal and pharmacological properties [92]. Larger terpenes exist as waxes and resins, as well as oxygenated terpenoids [93].

Kikuchi et al. [94] developed MIP-QCMs for the detection of terpenes. Poly-MAA MIPs (and NIPs) for α-pinene, limonene, and limonene-oxide were developed by drop-casting the oligomeric solution onto 10 MHz QCMs; 2,2-azobisisobutyronitrile (AIBN) was used as an initiator in this case. To remove the templates, the QCMs were washed in methanol (15 wt%) and acetic acid (0.5 wt%). The sensors exposed to limonene, limonene oxide, and a-pinene at 10 ppm of concentration exhibited larger shifts in frequency for the target gas. The MIP-QCM response for limonene oxide was higher than the other 2 terpenes-based MIP-QCMs, this was attributed to oxygen-related interaction among polymer and template. 

The detection of terpenes was also reported by Iqbal et al. [95], using a similar approach. A six channels array of 10 MHz QCMs was coated with MIPs prepared using terpenes typical of fresh and dried species of the Lamiaceae family, as rosemary, basil, and sage. MIPs were synthesized using styrene as functional monomer, DVB as cross-linker, AIBN as radical initiator, and diphenylmethane as porogen. α-pinene, limonene, eucalyptol, β-pinene, terpinene, and estragole were used as templates. The selectivity tested at 50 ppm concentration for each terpene demonstrated that each sensor was selective for the template used and that the array was able to discriminate all the molecules including isomeric compounds as α-pinene and β-pinene. 

PolyMAA MIPs-QCMs for terpenes (α-pinene, γ-terpinene and limonene) were also used by Hawari et al. [96] considering the aromatic volatile compounds emitted by Harumanis mango (a popular green eating mango variety which has been planted commercially in the State of Perlis, Malaysia). The highest response (in Hz) was obtained for α-pinene followed by γ-terpinene and limonene. The limonene MIP-QCM sensor was not selective in their array. Still on terpenes originated by mango Ghatak et al., [97] developed similar MIP-QCM for a marker of *Mangifera indica* L variety. The GS detected 3-carene in the 5 ppm to 1000 ppm range, with a selectivity factor of 91%.

Wang et al. [78] realised a terpenes MIPs-based QCM gas sensor array potentially useful for discrimination of P. orientalis trunks infested by the two insects *Symmorphus bifasciatus* and *Phloeosinus aubei*. GC-MS analysis of infected plants was used to identify four markers VOCs (α-pinene, β-phellandrene, 3-carene and *cis*-thujopsene) as molecular templates. The MIPs and NIPs were PMAA based. The NIPs coated QCM gold electrode had slight responses towards α-pinene, β-phellandrene, 3-carene, *cis*-thujopsene, D-limonene, *p*-cymene, γ-terpinene and hexanal, and no selectivity to these analytes compared with other MIPs-based QCM gas sensors. The four MIPs-QCMs were selective for the template and able to work in a 1–80 ppm range, demonstrating stability for at least one month.

Jha et al. [98] prepared three polyacrylic acid (PAA)-based MIP films using three VOCs (propenoic acid, hexanoic acid and octanoic acid) as templates on QCMs. The polymeric films were coated onto the surface of QCM by spin coating. The authors tested the array with different amounts of the single compounds and with binary mixtures of the organic acids. The average response (Hz) of the sensors to propenoic acid vapours was higher than the other two VOCs. For the binary mixtures, the highest average response was for propenoic acid + octanoic acid combination. The same research group [99] attempted a similar strategy for the identification of hexanal, heptanal, and nonanal in single, binary, and tertiary mixtures independently, and simultaneously in presence of humidity as the main interferent. The average sensor response (Hz) was maximum for the sensor with the heptanal MIP. The sensors’ performance was evaluated based on sensitivity, response time, and their ability to discriminate between different aldehydes using the pattern recognition methods: PCA and support vector machine (SVM). PCA results showed fair discrimination of binary, and tertiary mixtures further validated using the SVM classifier analysis of PC score matrices.

#### Applications in Real Samples

Ghatak et al. [100], following the work of Jha and Hayashi [99], developed a GS for furaneol, one of the flavor-enhancing aromas of mango, considering that furaneol flavor in mango vanishes during treatment of the fruit with calcium carbide (CaC_2_) for artificial ripening. Two groups of three different mango varieties (Amrapaly, Himsagar, and Langda) were studied; naturally ripened and CaC_2_ treated mango. When the headspace of the samples was analysed, the sensors had different frequency shifts for naturally ripened fruits; the signal was not significant for the samples treated with carbide. This was attributed to the lack of furaneol.

A very interesting approach for the discrimination of different essential oils from ginger (*Zingimber officinale* Roscoe) was realised by Hardoyono et al. [101]. PAA-MIPs on 9 MHz QCMs were prepared (by drop-casting) using as template borneol, citral, and geraniol aromas contained in essential oils from different varieties of ginger (i.e., *Zingiber officinale* var. amarum, SWG; *Zingiber officinale* var. officinale, BWG; *Zingiber officinale* var. rubrum, RG). Samples were firstly analysed using solid-phase micro-extraction (SPME)/GC-MS. Different amounts of the bioactive compounds were present. The data, analysed using principal component analysis (PCA) and linear discriminant analysis (LDA) indicated different groups of samples (SWG, BWG, and RG) distinguished from negative controls. Identification of borneol, citral, and geraniol using a MIPs-QCMs sensor array can, thus, represent a useful tool to identify the major components in essential oils and herbal samples.

An overview of the different MIP-based gas sensors’ main features and applications is reported in Table 2.

**Table 2 sensors-20-04433-t002:** Gas sensors equipped with Molecular Imprinted Polymers: main features and strategies.

Molecular Imprinted Polymers	Analyte	Transducer	Immobilization Technique	VOCs Concentration Measured	Application	Ref
MAA-MIP	Formaldehyde	QCM	Micro-syringe	≤ 2 ppm	-	[89]
MAA-MIP	Formaldehyde	QCM	Spin-coating	1–100 ppm	-	[90]
PMMA-MIP	Toluene, *p*-xylene	QCM	Spin-coating	Toluene 540 ppm; p-xylene 170 ppm	-	[91]
MAA-MIP	α-Pinene, limonene, limonene oxide	QCM	SAM	10 ppm	-	[94]
PDMS-MIP	α-Pinene, limonene, eucalyptol, β-pinene, terpinene, estragole	QCM	Spin-coating	50 ppm	Fresh herb	[95]
MAA-MIP	α-Pinene, γ-terpinene, limonene	QCM	Spin-coating	-	Harumanis mango	[96]
MAA-MIP	3-Carene, Furaneol	QCM	Drop-casting	5–1000 ppm	*Mangifera indica* var.: Langda, Amrapaly, Himsagar	[97,100]
MAA-MIP	α-Pinene, β-phellandrene, 3-carene, *cis*-thujopsene	QCM	Drop-casting	25 ppm	*Symmorphus bifasciatus* and *Phloeosinus aubei*	[78]
PAA-MIP	Propenoic acid, hexanoic acid, octanoic acid	QCM	Spin-coating	-	-	[98]
PAA-MIP	Propenoic acid, hexanoic acid, octanoic acid hexanal, heptanal, nonanal	QCM	Spin-coating	-	-	[99]
PAA-MIP	Borneol, neral, geraniol, citral	QCM	Drop-casting	-	*Zingiber officinale* var. amarum; *Zingiber officinale* var. officinale, *Zingiber officinale* var. rubrum	[101]

### 3.3. DNA

Nucleotidic acid sequences have been extensively used in sensors design, fabrication and application, in the past decade, providing new impulses to analytical research [102]. DNA sequences and structures, rationally designed, enable high-affinity interactions with a wide range of ligands, including vapor-phase odorants [103]. Chao et al. [104] briefly reviewed the history of DNA nanotechnology, summarizing the recent progress in DNA nanotechnology-based biosensors including the mechanisms and discussing the prospective development of DNA nanotechnology for the design and fabrication of advanced biosensors. The large majority of DNA-based sensors are designed to work in liquids [105]. The use of oligonucleotide sequences in gas sensing is still limited. However, as in the case of peptides, DNA in single-stranded structures can give rise to many combinations, offering modulable interaction. Folding of single-stranded structures, as for aptamers, and interaction with other nucleic acids chain (i.e., double-stranded, triple-stranded structures), allows targeting the desired interactions. Figure 4 shows examples of hairpin DNA structures.

One of the first papers in which DNA was used in gas sensing involved the fabrication of silver nanowires with a DNA template. In this work, Zhao et al. [103] used chemical reduction to fabricate Ag nanowires with DNA as a template (the DNA sequence was not reported). DNA was placed in a solution with AgNO_3_; the Ag^+^ coordinates with the negatively charged phosphate groups of DNA. The reduction bath induces the formation of metallic Ag nano-clusters on DNA molecules. The DNA-templated silver nanowires were deposited onto a gold interdigitated electrode to realise the sensors. Ammonia, hydrogen, ethanol, methanol, and acetone were tested and only ammonia gave a response. The electronic properties of the sensing materials are changed with the adsorption of gas molecules and change in conductance. No explanation about selectivity is reported. 

Shi et al. [106] proposed, for the first time, DNA for the development of organic field-effect transistor (OFET) for the detection of NO_2_. Genomic DNA from fish sperm was spray-coated to deposit DNA interlayer on the sensors. FETs without DNA were used as reference. NO_2_ ranging from 10 to 50 ppm was tested. The rapid increase in the response for the DNA OFET suggested that the sensing performance was higher compared to the reference FET. No other volatile compound was tested. 

Carbon nanotubes decorated with DNA have been also used as sensing element for the detection of volatile organic compounds. Khamis et al. [107] realised carbon nanotubes-based FETs decorated with DNA with favorable properties for gas sensing as rapid response, full recovery of the baseline (seconds), sensitivity towards many common odorants at low concentration (from ppb to ppm level). Single-walled carbon nanotubes (SWCNTs) grown by catalytic chemical vapor deposition on a SiO_2_/Si substrate were used. The FETs were constituted by Cr/Au source and drain electrodes patterned using optical lithography. A doped silicon substrate was used as a back gate. The oligonucleotides sequences were selected based on a precedent paper [108]. The 16 DNA sequences tested (Seq1, Seq2, Seq3, Seq4, and Seq5 having 21 bases; Seq6, Seq7, Seq8, Seq9. Seq10, Seq11, and Seq12 having 24 bases; Seq13, Seq14, Seq15, and Seq16 having 21 bases; see the abbreviation list at the end of the manuscript) were adsorbed onto the sensors by drop-casting. The mechanism underlying changes in SWCNTs conduction is supposed to be mainly because of electrostatic coupling among single-stranded DNA and the volatiles. Binding the odorant molecules indicates alteration of the conductance of the SWCNTs. DNA-SWCNTs resulted able to discriminate closely related analytes, for example among the homologous carboxylic acids: propanoic acid, hexanoic acid, and octanoic acid. Octanal, nonanal, and decanal were also discriminated and the response was correlated to the solubility of each compound in water. DNA-SWCNTs based on Seq1 was also able to differentiate between limonene and carvone enantiomers.

Kybert et al., [109] developed an array of vapor sensors consisting of CNT field-effect transistors functionalized with single-stranded DNA (DNA-SWCNTs). For the fabrication of this device, the electrical contacts for FETs with channels 10 μm long and 25 μm wide were patterned by photolithography and metalized with Cr/Au via thermal evaporation. Semiconducting SWCNTs were deposited by drop-casting onto the surface of the chip and the DNA in water was added. Four different DNA oligomers were used in this work: Seq17 with 21 bases, Seq18, Seq19, and Seq20 with 24 bases (see list of abbreviations). No indication about the selection of the sequences is given in the paper. The sensors reacted to dimethyl sulfone and isovaleric acid at concentrations ranging from 0.05 to 4 ppm. The responses were rapid and reversible. The DNA-SWCNTs sensors were also able to differentiate analytes with very similar molecular structure: limonene and three isomers of pinene, that has two structural isomers, each of which has a pair of enantiomers. The response of DNA-SWCNTs based on Seq17 to the enantiomers of limonene demonstrated clear discrimination between these highly similar molecules, also the isomers of pinene (at 130 ppm) were clearly distinguished by DNA-SWCNTs.

The possibility of using oligonucleotides with a particular hairpin conformation (for VOC-detection) has recently been demonstrated. Hairpin DNA (hpDNA) is composed of a stem (used to orientate the immobilization onto the sensor) and a loop, that can interact with the VOCs. The loop size and nucleotidic sequence allow modulation of the interaction. 

In this respect, Mascini et al. [110] developed a gas sensor array with different hpDNAs conjugated with AuNPs onto 20 MHz QCMs modified via drop-casting. The purified oligonucleotides also had a thiol spacer having six carbons (C6) attached to 5’ phosphate end of the hpDNA for the binding with AuNPs. The hpDNAs loop binding ability was calculated in silico. Eight different VOCs (ethanol, 3-methylbutan-1-ol, 1-pentanol, octanal, nonanal, ethyl acetate, ethyl octanoate, and butane-2,3-dione) were tested using seven different sequences selected for their different binding scores. Sequences with the loop of different sized were employed (tetramer loop, pentamer loop, and hexamer loop): HpDNA3, HpDNA4, HpDNA6, HpDNA7, HpDNA8, HpDNA9, HpDNA10 (see the abbreviations list at the end of the manuscript). The piezoelectric sensorgram obtained was similar for all hpDNA-AuNP and VOCs, showing a rapid decrease of the signal when the target analyte was sent into the e-nose chamber; followed by a slower raise up to the steady-state. The calculations of the binding constants for the eight VOCs resulted in a very similar binding affinity for the tetrameric loops. The tetramer HpDNA10 exhibited a slightly better affinity for aldehydes leading to a significant correlation with simulated results. 1-pentanol and 3-methylbutan-1-ol resulted to be bound by the pentamer HpDNA6. The other DNA pentamer loop, HpDNA4, had the lowest binding affinity for all molecules. Hexameric loops HpDNA8 and HpDNA7 showed a significant interaction with ligands, which was approximately two-fold higher than the smaller DNA loop. These results were in good agreement with the prediction by virtual screening.

Gaggiotti et al. [111] reported also the realization of an optoelectronic nose, where the GSs array was composed of the above-mentioned hairpin DNA sequences in combination with peptides selected by the virtual screening [69]. The authors evaluated the performance of the SPRi optoelectronic nose for the analysis of different VOCs: 1-butanol, 1-pentanol, 1-hexanal, 1-nonanal, *trans*-2-nonenal, and 1-hexanoic acid. Six sequences of peptides (Ps18, Ps19, Ps20, Ps21, Ps22, Ps231), three sequences of hpDNA with unpaired tetramer loops (HpDNA1, HpDNA2, HpDNA3), three pentamers (HpDNA4, HpDNA5, HpDNA6), and three hexamers (HpDNA7, HpDNA8, HpDNA9) loops were spotted onto the prisms. The dataset obtained in real-time was normalized and elaborate with PCA and Hierarchical Clustering (AHC) analysis. Different reactivity pattern was noticed for peptides and hpDNAs with peptides having larger discriminating ability than DNA. The AHC analysis allowed to discriminate very similar molecules (VOCs of the same family with only 1-carbon difference) demonstrating that peptides and hpDNA contributes synergistically to the property of the array. The experiments with all VOCs tested were conducted over two weeks repeatability was good (inter-day CVs 10–15%).

#### Applications in Real Samples

To the best of our knowledge, only two applications on real samples have been reported for DNA-based GSs in carrot and hemp samples. The same sensor array developed in [110] was used by Gaggiotti et al. [112] to monitor volatiles profile change in carrots. A single batch of fresh carrots was blanched to inactivate enzymes and stored at different temperatures for 26 days. The headspace of the samples was analysed by SPME/GC-MS and the GSs array in parallel. The most represented volatile compounds α-pinene and γ-terpinolene were detected in all samples stored at the different temperatures, except for samples stored at 40 °C. Butane-2,3-diol, acetoin and lactamide, were the volatiles present in storage at 25 °C from the 8th day and during storage at 40 °C. Fermentation during the storage was supposed because of the presence of acetoin and butane-2,3-diol. The data of the hpDNA array analysed using PCA, were very similar to SPME/GC-MS. This study demonstrated, for the first time, the ability of a hpDNA based gas sensor array to evaluate volatile organic compounds headspace in solid food matrices.

A second application was related to the comparison of the performance of hpDNA and peptide QCMs arrays for the detection of terpenes of hemp samples [113]. In this case, the sensor array was composed of Ps18, Ps19, Ps20, Ps21, Ps22, Ps23, and HpDNA1, HpDNA6, HpDNA4, HpDNA8, HpDNA. The terpenes volatile fraction in different hemp samples purchased from three Italian regions was initially determined using SPME/GC-MS to classify the samples. Samples were classified into two groups. Among the different VOCs identified there were monoterpenes (19 in total). Nine sesquiterpenes included one alcohol, one epoxide, and three aromatics. A Pearson coefficients analysis gave a higher correlation for hpDNA than the peptides toward the VOCs significant to classify the samples. HpDNA6 and HpDNA4 were anticorrelated with β-caryophyllene, L-borneol, and α-terpineol. A positive correlation was observed for HpDNA8 and *p*-mentha-8-thiol-3-one and HpDNA7 with both L-borneol and α-terpineol. These two alcohols correlated also with the peptide Ps19 that was the only peptide showing a significant correlation with the VOCs significant to classify the hemp samples. 

Using PCA a similar recognition performance was observed for all the peptides. On the other hand, hpDNA loops played an important role in the separation of the hemp samples. These data demonstrate the different interaction ability of hpDNA and peptides for terpenes. The authors concluded that a mixed set of binding elements can provide a synergistic response in the detection of VOCs. Table 3 provides an overview of the different DNA-based gas sensors’ main features and applications.

**Table 3 sensors-20-04433-t003:** Gas sensors equipped with DNA sequences: main features and strategies.

DNA	Analyte/Samples	Transducer	Immobilization Technique	Concentration of VOCs	Application	Ref
Ag nanowires DNA-template	Ammonia	Gold interdigitate electrode	-	200 ppm	-	[103]
DNA-fish	NO_2_	FET		10–50 ppm		[106]
DNA-SWCNTs	Propanoic acid, hexanoic acid, octanoic acid	FET	-	Propanoic acid 1100 ppm: hexanoic acid 1100 ppm: octanoic acid 790 ppm; limonene 0.3–1500 ppm: 0.05-carvone 250 ppm	-	[107]
DNA-SWCNT	Dimethyl-sulfone, isovaleric acid α-pinene β-pinene	FET	-	Dimethyl-sulfone and isovaleric acid 0.05–0.4 ppm; pinene 130 ppm	-	[109]
HpDNA3, HpDNA4, HpDNA6, HpDNA7, HpDNA8, HpDNA9, HpDNA10	Ethanol, 3-methylbutan-1-ol, 1-pentanol, octanal, nonanal, ethyl acetate, ethyl octanoate, butane-2,3-dione	QCM	Drop casting	-	-	[110]
HpDNA1, HpDNA2, HpDNA3, HpDNA4, HpDNA5, HpDNA6, HpDNA7, HpDNA8, HpDNA9	1-Butanol, 1-pentanol, 1-hexanal, 1-nonanal, trans-2-nonenal and 1-hexanoic acid	SPRi	Micro spotting robot	1-butanol 55 ppm; 1-pentanol 31 ppm; 1-hexanal 90 ppm; 1-nonanal 4 ppm; trans-2-nonenal 7 ppm	-	[111]
HpDNA1, HpDNA2, HpDNA3, HpDNA4, HpDNA5, HpDNA6, HpDNA7, HpDNA8, HpDNA9	Terpenes, alcohol, aldehydes, ketones	QCM	Drop casting	-	Fresh carrots	[112]
HpDNA1, HpDNA6, HpDNA4, HpDNA8, HpDNA7	Terpenes	QCM	Drop casting	-	*Cannabis sativa* L.	[113]

## 4. Conclusions

This review aimed to provide a state of the art in the development of GSs based on peptides, DNA and MIPs, paying particular attention to their applications. Looking at the reported papers we can say that remarkable activities have been reported for all the three selected sensing elements.

The use of peptides as gas sensing elements is in a more advanced status since they represent the “natural” extension of the use of olfactory receptors. Moreover, they have greater variability of the response towards different VOCs since larger different potential combinations of amino acids in the final sequence. In this respect, many efforts have been devoted to searching for affinity for a particular volatile or a class of volatiles taking advantage of the possibility to predict the interactions “in silico”. The studies generally confirmed affinity (and selectivity) for the targets. Particularly significant appears the use of array of peptides having different binding ability to discriminate very similar molecules (i.e., among series of organic acids or enantiomers of terpenes). These very relevant data can be achieved using either a computationally designed library or a whole set of peptides having different behaviors. It should be also noticed that oligopeptides are very flexible and have been used in conjunction with all the transducers reported, thus, are useful for devices having different sensitivity. Most of the works take advantage of the formation of SAM onto the sensor surface by using a cysteinyl end residue, in some cases immobilization on nanomaterials is carried out to improve the number of binding sites and maximizing the binding element exposure to VOCs. 

The use of MIPs in GSs gave interesting data as well; in the majority of the papers, the MIP GSs were reported to have very high selectivity (i.e., for enantiomeric terpenes) as for MIPs used in solution either on sensors or in microextraction. All the developed sensors used QCMs as transducer realising the polymer directly onto the sensor surface. This is probably due to the major limitation that is represented by the polymerization step; in fact, it is hard to control the thickness of the film onto the sensor surface to have a working transducer and a number of binding sites able to measure the volatiles. In this respect, QCMs appear a straightforward approach since it measures directly the “mass” adsorbed onto the microbalance. Few attempts have been made to develop GSs arrays, none to use different polymers for the same volatile pattern, neither the rationalization of the binding in silico.

The use of DNA in gas sensing is more recent and should be still explored in more detail. Few attempts have been done using the “natural” property of the double helix to give recognition because of the conformation of the double-strand. Both stabilization and transducing ability have been given in conjunction with metal or carbon nanomaterials. The use of particular sequences to develop arrays has given interesting results for different classes of VOCs. Despite the lower intrinsic variability of DNA sequence with respect to peptides, recent works demonstrated that a synergistic effect can be achieved using mixed peptide-DNA arrays.

The stability of these gas sensors in the long-term was examined by few authors. The lifetime of these sensors depends on the sensitive element, the method of deposition on the surface of the sensor, and their use in terms of number of measures. Some authors such as Mascini et al. [70] tested the stability of peptide-based sensors on different days of analysis, Wang et al. [78] tested the stability of the sensors, based on MIPs, exposing them every 5 days to known concentrations of target gas up to a maximum of 30 days obtaining the same response, Gaggiotti et al. [112] tested the reproducibility of the hpDNA-based sensor array using octanal as the target gas over three months. By these few examples we can affirm that gas sensors based on peptides, MIPs and DNA appear stable enough to be used for robust detection in the long-term. However, more studies are necessary to assess the effective long-term stability for measurements on real samples in continuous use. 

The target gases analysed by the sensors are VOCs (apart from ammonia and nitrogen dioxide) that are potentially useful in many applications: health, food, environment, process control in industry, traceability etc. However, the application on real samples are still limited, very few on breath-test and mostly on food. This represents a crucial point for the validation of the GSs realized and for the possibility to introduce devices based on this kind of sensors in the market. It should be pointed out that applying the GSs to real cases is very challenging; in fact, sampling should be designed, a parallel analysis for the identification and quantification of the analytes is necessary (often by GC-MS), and a very high number of samples is mandatory to build and validate the model for multivariate data analysis. 

Looking in a perspective way to this field of research, it is possible to see potential improvement in the performance of this GSs, the recent introduction of a portable SPRi, may give further input to the potential application of both peptide-based and DNA-based GS arrays since the same chemistry can be used to realize the arrays. The use of the virtual libraries already developed or the calculated affinity different classes of VOCs may help in the realizations of arrays used for specific applications (i.e., the volatile pattern of breath is different from foods) improving the performance of the analysis and reducing the complexity of the statistical treatment of the data. The mixed approach (different types of binding elements) can further help in this respect. 

The use of MIPs appears more complex and should be better investigated, particularly for the realization of arrays and for their possible use in conjunction with transducers others than QCMs. However, considering their large use in liquid phase and the recently developed methods to obtain nanodispersions or for the synthesis of thin-films, there is room, in our opinion, for their use in GSs and GS arrays since they can provide selectivity and robustness.

This review demonstrates that peptides, MIPs and DNA can be successfully used in GSs and GS arrays and have great potential to be binding elements of election for the development of GSs and GS-arrays for future applications. More studies directed to improve the performance of the array are needed particularly in real cases. In conclusion, we can affirm that these sensors can play a crucial role in the analysis of real samples in different fields such as medicine, food quality, or environmental analysis. Moreover, their use in combination with emerging technologies as microengineering, nanotechnologies, and advanced rational design will certainly bring resolving further challenging analytical and real-life issues in the near future.

## Figures and Tables

**Figure 1 sensors-20-04433-f001:**
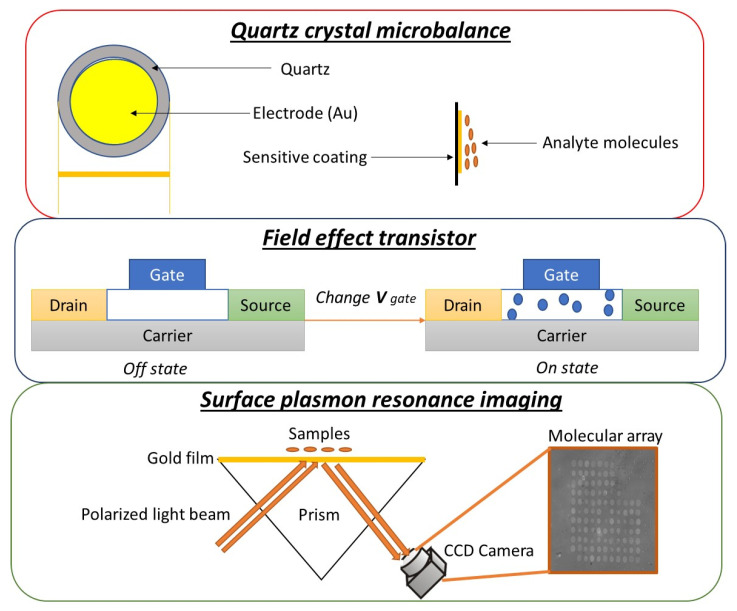
Sketch of the most used transducers in peptides, DNA and MIPs modified gas sensors.

**Figure 2 sensors-20-04433-f002:**
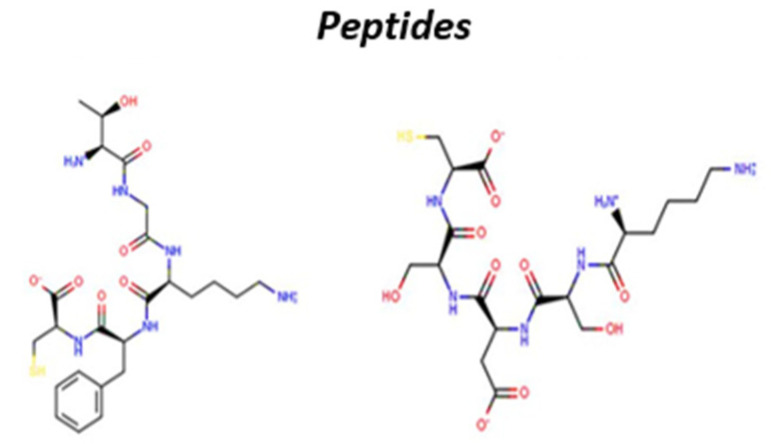
Sketches of oligopeptides.

**Figure 3 sensors-20-04433-f003:**
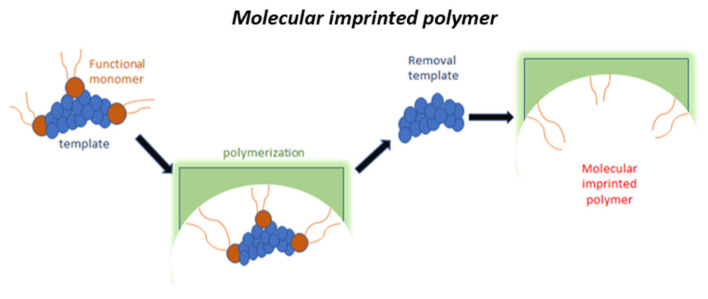
Graphical scheme of MIPs assembling.

**Figure 4 sensors-20-04433-f004:**
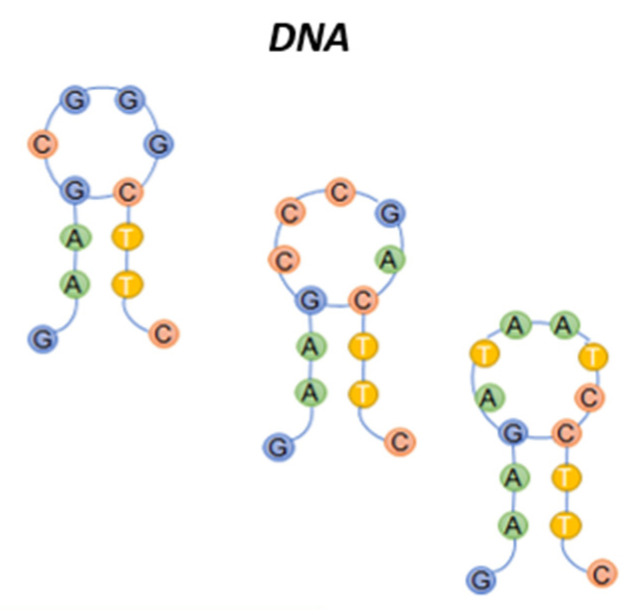
Sketch of Hairpin DNA structures with different loop dimensions.

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
