# Peer review of "Peptides, DNA and MIPs in Gas Sensing. From the Realization of the Sensors to Sample Analysis"

_sensors, 2020, doi:10.3390/s20164433_

Round 1

Reviewer 1 Report

The contents of this paper should be very interesting to readers. It gives a review of the three emerging biosensor techniques which includes peptide, molecularly imprinted polymers and DNA based gas sensors/ arrays. The only fly in the ointment is as follows:

  • In the tables of the paper, please pay attention to seperate the items, for example, in Table 1, people cannot recognize the bound between Analyte of PS2 and PS7, PS8. The same problem lies in Table 2 and Table 3.
  • For the three kinds of gas sensors, how about their lifetime, their resuability? These are very important points. It will be better to give readers these information.
  • Some clerical errors or phrase, which are hard to understand, need to be corrected:

Line 102, “Among these analytical requirements result very important have a minimum or no sensor drift”;

Line 192, “Each section reports at the end a table (Table 1......”

Line 474, “The selectivity of the MIPs-QCM vs. the porogen solvent used was achieved.”

Line 700, “Since multivariate statistic is applied a crucial role is also played......”

Line 717, “propriety” shoudl be “property”?

Author Response

Reviewer 1

The contents of this paper should be very interesting to readers. It gives a review of the three emerging biosensor techniques which includes peptide, molecularly imprinted polymers and DNA based gas sensors/ arrays. The only fly in the ointment is as follows:

We wish to thank the referee for the review and for appreciating our work.

In the tables of the paper, please pay attention to seperate the items, for example, in Table 1, people cannot recognize the bound between Analyte of PS2 and PS7, PS8. The same problem lies in Table 2 and Table 3.

Thanks for the suggestion. The tables 1, 2, and 3 have been modified as suggested

For the three kinds of gas sensors, how about their lifetime, their reusability? These are very important points. It will be better to give readers these information.

 Thank you for the suggestion, we have introduced in the conclusions section the following sentences:

The stability of these gas sensors in the long-term was examined by few authors. The lifetime of these sensors depends on the sensitive element, the method of deposition on the surface of sensor, and their use in terms of number of analysis.  Some authors such as Mascini et al. [71] tested the stability of peptide-based sensors on different days of analysis, Wang et al. [79] tested the stability of the sensors, based on MIPs, exposing them every 5 days to known concentrations of target gas up to a maximum of 30 days obtaining the same response, Gaggiotti et al. [82] tested the reproducibility of the hpDNA-based sensor array using octanal as the target gas over three months. By this few examples we can affirm that gas sensors based on peptides, MIPs and DNA appears stable enough to be used for a robust detection in long-term. However, more studies are necessary to assess the effective long-term stability for measurements on real samples in continuous use.   

Some clerical errors or phrase, which are hard to understand, need to be corrected:

Line 102, “Among these analytical requirements result very important have a minimum or no sensor drift”;

Changed in “As already reported in the introduction is very relevant to reduce sensor drift”

Line 192, “Each section reports at the end a table (Table 1......”

Changed in “A table (Table 1-3) summarizing the main features of the sensors in terms of the binding element, analyte/sample, transducer, immobilization strategies, VOCs detected amount is reported at the end of each paragraph”.

3.1 Peptides

Line 474, “The selectivity of the MIPs-QCM vs. the porogen solvent used was achieved.”

Changed in “Using this approach selectivity of the MIPs-QCM sensors  for the porogen solvent used to synthesize the polymer was achieved.”

Line 700, “Since multivariate statistic is applied a crucial role is also played......”

The sentence has been eliminated

Line 717, “propriety” shoudl be “property”?

Changed as requested

Reviewer 2 Report

This is well written manuscript and should be published. I believe that many readers will read it because of the review of recent and novel gas sensing methods.

I have a few remarks which should be amended before publication:

  1. l. 56: The issue of drifts was considered by many authors and you should add some references and remarks about the differential methods used to reduce these inadvisable effects.
  2. l. 80: You have mentioned PCA DA and CA methods. It would be valuable to add explanation for DA method. Two words "probabilistic analysis" is insufficient.
  3. l. 111: The presented FET sensors are are limited to MOS FET sensors. You should mention about the FET made of two-dimensional materials, presented at e.g., Rumyantsev, S., Liu, G., Shur, M. S., Potyrailo, R. A., & Balandin, A. A. (2012). Selective gas sensing with a single pristine graphene transistor. Nano letters, 12(5), 2294-2298. Such FET sensors, as well as MOS sensors, utilize 1/f noise phenomena and statistical analysis to determine the components of gas mixtures. You should mention about other statistical methods of gas detection, e.g., Smulko, J. M., & Kish, L. B. (2004). High-order statistics for fluctuation-enhanced gas sensing. Sensors and Materials, 16(6), 291-299.
  4. l. 263 You have mentioned an opto-electronic nose based on plasmon resonance. You should add 1-2 sentences presenting repeatibility of this method and the main principles of optical set-up.
  5. l. 361 the mentioned MDA method should be explained by adding 1-2 sentences.
  6. l. 471, please use the space in "4 MHz" after the number.
  7. You don't use subscripts in the names of chemical compounds, e.g., l. 526 CaC2, l. 568 AgNO3, l. 580 NO2

Author Response

This is well written manuscript and should be published. I believe that many readers will read it because of the review of recent and novel gas sensing methods.

I have a few remarks which should be amended before publication:

We wish to thank the referee for the review and for appreciating our work.

  1. 56: The issue of drifts was considered by many authors and you should add some references and remarks about the differential methods used to reduce these inadvisable effects.

Some recent information on treatment of the drift has been given in the introduction as follows:

Furthermore, in order to reduce the drift effect in the GSs, new algorithms have been developed. Some approaches use multivariate analysis such as principal component analysis (PCA, see later) to compensate for drift [11-13]. Liu et al. [14] , interestingly, used an active learning (AL) methodology, which allowed to solve the sensors drift. This method can select a certain number of incoming samples and update the classifiers reliably.

  1. 80: You have mentioned PCA DA and CA methods. It would be valuable to add explanation for DA method. Two words "probabilistic analysis" is insufficient.

As suggested by the reviewer, description has been improved introducing the following part: 

DA is an explanatory and predictive analysis. Two different models of DA can be used to obtain different informations: a) to which group an observation will belong to b) if the groups to which the observations belong are distinct. A linear model can be used if the covariance matrices are assumed to be identical, while a quadratic model is used when the covariance matrices differ in at least two groups [24,25]. CA is used to split data into groups depending on the similarities or distances between points in the data set.  Other methods that allows to study the results obtained from gas sensors are  high-order statistics (HOS). These methods allow the extraction of more information than signal analysis and may lead to significant improvements in selectivity and sensitivity of sensor responses [26]

  1. 111: The presented FET sensors are are limited to MOS FET sensors. You should mention about the FET made of two-dimensional materials, presented at e.g., Rumyantsev, S., Liu, G., Shur, M. S., Potyrailo, R. A., & Balandin, A. A. (2012). Selective gas sensing with a single pristine graphene transistor. Nano letters, 12(5), 2294-2298. Such FET sensors, as well as MOS sensors, utilize 1/f noise phenomena and statistical analysis to determine the components of gas mixtures. You should mention about other statistical methods of gas detection, e.g., Smulko, J. M., & Kish, L. B. (2004). High-order statistics for fluctuation-enhanced gas sensing. Sensors and Materials, 16(6), 291-299.

The question has been partially answered before.  We wish to point out  that  the description of the FETs was not intended to be ehaustive; it was thought as a brief explanation of the transducers already used in conjunction with peptides, MIPs and DNA. The following sentence was inserted for clarity as suggested by the reviewer:

However, another type of material can be used to modify the FET’s gate, graphene. It has been shown that these sensors can have different strengths such as high electrical conductivity, surface-volume ratio, low thermal resistance and relatively low 1/f noise and the ability to strongly tune the conductivity of the gate. These aspects make these sensors promising for gas detection applications [34]; at the moment no application on peptides, MIPs or DNA sensors has been reported.

  1. 263 You have mentioned an opto-electronic nose based on plasmon resonance. You should add 1-2 sentences presenting repeatibility of this method and the main principles of optical set-up.

The following sentences were introduced:

“ SPRi technology allows measurements using a fixed refraction angle called the working angle θw. Using an angle of incidence at the maximum slope of the plasmon curves, changes in resonance conditions can produce large variations in the intensity of reflectivity. In this system, the authors used a 16-bit video camera (CCD) to capture the interaction images between the VOCs and the sensor array.

 About  repeatability, this sentence was added at the end of the paragraph related to ref 111

The experiments with all VOCs tested were conducted over two weeks repeatability was good ( inter-day CVs 10-15%).  

  1. 361 the mentioned MDA method should be explained by adding 1-2 sentences.

The following sentence has been added:

that is a multivariate dimensionality reduction technique. This method allows objects to be allocated consistently and most appropriately to groups for which representatives have already been selected

  1. 471, please use the space in "4 MHz" after the number.

 The space have been included.

You don't use subscripts in the names of chemical compounds, e.g., l. 526 CaC2, l. 568 AgNO3, l. 580 NO2

 The chemical compounds have been modified.

Reviewer 3 Report

This paper is relatively well written and comprehensive in coverage of this specialized subject pertaining to the detection of VOCs using peptide, DNA, & MIPs in gas sensors. The organization is fine and the figures are well presented. However, there are some English grammar and sentence structural changes (awkward, unclear or confusing wording) that require revisions and clarifications as I will outline in the following examples.

Abstract rewording

L12-19 - ….as well as in 'point-of-care' diagnostic testing. Gas chromatography coupled with instruments having various sensors remains the reference method for analysis of volatile organic compounds (VOCs), but gas sensors are now available to provide additional analysis capabilities, applications for problem solving, and new simpler (easier to use) tools for VOC detection.

Introduction

L29-30 ….and complex human senses. Particular smells can trigger peoples' memories and provide information useful in self defense and personal safety. The discovery of genes encoding...

2. Transducers

L111 - Field-effect transistors (FETs) are devices...[no not bold or use colon]

L126 - Piezoelectric sensors with vibrating crystals are used in different fields, ...[no not bold or use colon]

L145 - Surface Plasmon Resonance (SPR) is used extensively for… [do not bold or use colon]

Do not begin a new paragraph with a dependent clause or preposition [instead, place all wording (before the comma) at the end of the sentence. This applies only to the first sentence in a new paragraph (as in the following cases):

L179

L190

L238 Lu et al. [55] in 2009 used a...342

L262

L275 Moho et al. [59] recently used...

L285 omit the preposition (not needed)  One of the attractive features …

L335 same correction as for L238

L342

L354 spell out 98  Ninety-six patients in...

L401 omit the preposition (not needed)

L456 same correction as L238

L524 Ghatak et al. [93], following the work of Jha and Hayashi [92], developed a GS for... [switch numbers and order for these two references (to correct proper sequence, so Ghatak et al. becomes ref [92], Jha ref [93])]

L548

L619 The possibility of using oligonucleotides with a particular hairpin conformation (for VOC-detection) has recently been demonstrated.

L623

L655

L657 move this sentence up to follow L656

References

All citations in the Reference section need to be reformatted using the (Paragraph: Indentation, Special: Hanging 0.3", Line spacing at least 13 pt and 0 pts before and after)-function in Word documents, after each number for references (using a tab),  which is built-in if you used the Sensors manuscript template available at the Authors website. This will align all words in each reference with a consistent 0.3" indent (on the left side).

Author Response

Reviewer 3

This paper is relatively well written and comprehensive in coverage of this specialized subject pertaining to the detection of VOCs using peptide, DNA, & MIPs in gas sensors. The organization is fine and the figures are well presented. However, there are some English grammar and sentence structural changes (awkward, unclear or confusing wording) that require revisions and clarifications as I will outline in the following examples.

We wish to thank the referee for the review and for appreciating our work.

Abstract rewording

L12-19 - ….as well as in 'point-of-care' diagnostic testing. Gas chromatography coupled with instruments having various sensors remains the reference method for analysis of volatile organic compounds (VOCs), but gas sensors are now available to provide additional analysis capabilities, applications for problem solving, and new simpler (easier to use) tools for VOC detection.

Thanks for the suggestion. The abstract have been modified in the manuscript.  The selected lines have been changed as follows:

“Detection and monitoring of volatiles is a challenging and fascinating issue in   environmental analysis, agriculture and food quality, process control in industry, as well as in 'point of care' diagnostic. Gas chromatographic approaches remains the reference method for the analysis of volatile organic compounds (VOCs); however, gas sensors (GSs) have become a reality with advantages of low cost and no or very little sample preparation. Gas sensors can be used singularly or in array format (e.g. e-noses); coupling data output with multivariate statical treatment allows un-target analysis of samples headspace”.

Introduction

L29-30 ….and complex human senses. Particular smells can trigger peoples' memories and provide information useful in self defense and personal safety. The discovery of genes encoding...

The sentence has been eliminated

Transducers

L111 - Field-effect transistors (FETs) are devices...[no not bold or use colon]

L126 - Piezoelectric sensors with vibrating crystals are used in different fields, ...[no not bold or use colon]

L145 - Surface Plasmon Resonance (SPR) is used extensively for… [do not bold or use colon]

 Modified as requested.

Do not begin a new paragraph with a dependent clause or preposition [instead, place all wording (before the comma) at the end of the sentence. This applies only to the first sentence in a new paragraph (as in the following cases):

L179

L190

L262

L342

L548

L623

L655

The sentences have been modified in the manuscript.

L238 Lu et al. [55] in 2009 used a...342

L275 Moho et al. [59] recently used...

L285 omit the preposition (not needed)  One of the attractive features …

L335 same correction as for L238

L354 spell out 98  Ninety-six patients in...

L401 omit the preposition (not needed)

L456 same correction as L238

L524 Ghatak et al. [93], following the work of Jha and Hayashi [92], developed a GS for... [switch numbers and order for these two references (to correct proper sequence, so Ghatak et al. becomes ref [92], Jha ref [93])]

L619 The possibility of using oligonucleotides with a particular hairpin conformation (for VOC-detection) has recently been demonstrated.

L657 move this sentence up to follow L656

 The manuscript has been corrected as requested

References

All citations in the Reference section need to be reformatted using the (Paragraph: Indentation, Special: Hanging 0.3", Line spacing at least 13 pt and 0 pts before and after)-function in Word documents, after each number for references (using a tab),  which is built-in if you used the Sensors manuscript template available at the Authors website. This will align all words in each reference with a consistent 0.3" indent (on the left side).

 References have been corrected.
